# Genetic testing offer for inherited neuromuscular diseases within the EURO-NMD reference network: A European survey study

Borut Peterlin[1], Francesca Gualandi[2☯], Ales Maver[1☯], Serenella Servidei[3], Silvère M. van der Maarel[4], Francoise Lamy[5], Alexander Mejat[5], Teresinha Evangelista[6]*, Alessandra Ferlini[2]*

1 Clinical Institute of Medical Genetics, University Medical Centre Ljubljana, Ljubljana, Slovenia, 2 Unit of Medical Genetics, University Hospital Ferrara, Ferrara, Italy, 3 Neurophysiopathology Unit, Fondazione Policlinico Universitario A. Gemelli IRCCS, Rome, Italy, 4 Department of Human Genetics, Leiden University Medical Center, Leiden, The Netherlands, 5 AFM-Téléthon, Evry Cedex, France, 6 Neuromuscular Morphology Unit, Myology Institute, GHU Pitié-Salpêtrière, Sorbonne Université, Paris, France

☯ These authors contributed equally to this work.
* fla@unife.it (AF); t.evangelista@institut-myologie.org (TE)

**Data Availability Statement:** All relevant data are within the manuscript and its Supporting Information files.

## Abstract

The genetic diagnostics of inherited neuromuscular diseases (NMDs) is challenging due to their clinical and genetic heterogeneity. We launched an online survey within the EURO-NMD European Reference Network (ERN) to collect information about the availability/distribution of genetic testing across 61 ERN health care providers (HCPs). A 17 items questionnaire was designed to address methods used, the number of genetic tests available, the clinical pathway to access genetic testing, the use of next-generation sequencing (NGS) and participation to quality assessment schemes (QAs). A remarkable number of HCPs (49%) offers $\geq$ 500 genetic tests per year, 43,6% offers 100–500 genetic tests per year, and 7,2% $\leq$ 100 per year. NGS is used by 94% of centres, Sanger sequencing by 84%, MLPA by 66% and Southern blotting by 36%. The majority of centres (60%) offer NGS for all patients that fulfil criteria for NMD of genetic origin. Pipelines for NGS vary amongst centres, even within the same national system. Referral of patients to genetic laboratories by specialists was frequently reported (58%), and 65% of centres participates in genetic testing QAs. We specifically evaluated how many centres cover SMA, DMD, Pompe, LGMDs, and TTR genes/diseases genetic diagnosis, since these rare diseases benefit from personalised therapies. We used the Orphanet EUGT numbers, provided by 82% of HCPs. SMA, DMD, LGMD, TTR and GAA genes are covered by EUGTs although with different numbers and modalities. The number of genetic tests for NMDs offered across HCPs National Health systems is quite high, including routine techniques and NGS. The number and type of tests offered and the clinical practices differ among centres. We provided evidence that survey tools might be useful to learn about the state-of-the-art of ERN health-related activities and to foster harmonisation and standardisation of the complex care for the rare disease patients in the EU.

**Funding:** The author(s) received no specific funding for this work.

**Competing interests:** The authors have declared that no competing interests exist.

## Introduction

Genetic neuromuscular disorders (NMDs) are individually rare. Yet, on the whole, they are rather frequent in Europe with an estimated prevalence of 1.6/100,000. Therefore, they represent a significant health burden to society [1]. NMDs involve skeletal muscle and often heart, peripheral nerves, neuromuscular junctions and spinal cord motor neurons. The vast majority has a genetic cause, with ≥600 genes already identified (see http://www.musclegenetable.fr/index.html; [2]). Due to the significant clinical and genetic heterogeneity, either for the many genes involved (genetic heterogeneity) and the great variety of mutation types in a single gene (allelic heterogeneity), NMDs are often challenging to diagnose.

Consequently, the diagnostic process may take several months or even years, require several specialists and need many medical investigations, including invasive ones (i.e. muscle biopsy). Genetic diagnosis is in our days recognised as mandatory for NMDs since it allows proper prevention, family planning, and access to therapy or novel clinical trials. This is even more true given the availability of personalised treatments; a prime example is the new drugs for spinal muscular atrophy (SMA) [3]. Moreover, not only individual diagnostic medical centres but also national health systems need to seek cross-border collaboration in terms of provision of comprehensive genetic testing. Different molecular biology techniques can be used to address the full mutational spectrum found in NMDs which includes: single nucleotide variants, large deletions and duplications (either as simple copy number variations (CNVs) or complex rearrangements), small mutations, epigenetic changes, dynamic mutations and atypical mutations or alterations occurring in regulatory regions as promoters, untranslated 5'/3' regions, or intergenic segments.

Recently, the diagnostic approach of rare genetic disorders, including NMD, has been revolutionized by the implementation of next-generation sequencing (NGS) in clinical settings. The main advantage of NGS is that any given number of genes can be tested in a single genetic test. Either an a priori selected list of several tens of genes (NGS panels) known to be associated with the type of disorder the patient is tested for, all the genes related to human diseases (clinical exome sequencing—CES), or even all genes in the genome (whole-exome sequencing– WES or whole genome sequencing -WGS).

When comparing with traditional genetic testing, there is cumulating evidence that NGS has superior diagnostic yield and cost-saving potential. This was shown for several groups of NMDs including inherited muscle disorders, motor neuron disorders, hereditary ataxias, hereditary motor and sensor neuropathies and hereditary spastic paraplegias [4–8].

However, the translation of novel genomic technologies into clinical practice has been slow and heterogeneous across European countries. This is mainly due to a lack of standards for implementation of genomic testing, limited access to expertise and testing, lack of institutional and clinician acceptance and lack of reimbursement [9]. The reimbursement strategy is not uniform across Europe as each member state has its own policies and reimbursement may be dependent of approval by either private or public insurance schemes, consequently, it might be expected that the level of implementation of genetic testing, including NGS, differs in different health systems across Europe.

The European reference network for rare neuromuscular disorders (ERN-EURO-NMD) launched a survey among members of the network to assess the current availability of genetic testing as well as the dissemination of NGS in the field of NMD amongst the member states.

With this work, we aim to provide evidence that survey tools, used in the context of the ERNs, are useful means to collect information about the state-of-the-art of health-related activities for rare diseases. Well-designed surveys help to increase knowledge about specific health themes, unmet needs, or forgotten niches often left unaddressed. The acquired knowledge will help to improve and homogenise health care service to the rare disease community.

## Materials and methods

EURO-NMD genetic task group constructed a 17 items, web-based questionnaire which addressed: characteristics of the participating centres (number of tests and Orphanet registration); technologies used for genetic testing (standard and NGS); the use of NGS in the clinical setting, including technical aspects (coverage and in-house bioinformatics), clinical indications, prioritisation and clinical pathway for NGS; approach to handling variants of unknown significance (VOUS) and reporting incidental findings; in house NGS algorithms; main barriers to the implementation of NGS, how these barriers may be overcome; and participation in external quality assessment schemes (S1 Table). The web-based survey was sent to the 61 healthcare providers of the ERN EURO-NMD in September 2018. The questionnaire remained open until February 2019. Data was analysed using descriptive statistics. This study did not require ethical approval since no human subjects or human biological material was used; the questionnaire was GDPR compliant.

## Results

### Responses

We received completed questionnaires from 55 out of 61 centres (response rate 90,2%) in 12 EU countries; 81.8% of the participating centres were registered in the Orphanet database as confirmed by their EUGT unique code (S2 Table). All centres were public institutions. Regarding the number of genetic tests performed per year the majority (24) performed between 100–500 genetic tests (43.6%), 13 centres (23.6%) performed between 500–1000 tests, 13 centres (23.6%) between 1000–5000, 4 centres (7,2%) less than 100 tests, and only one centre (1.8%) conducted more than 5000 genetic tests per year (Fig 1a). Therefore, the offer for genetic tests for rare NMDs occurs in 90.9% of HCPs participating to the survey (Fig 1b).

### NGS use and implementation

NGS is used by most participating centres (94%), followed by Sanger sequencing (84%), MLPA (66%) and Southern blotting (36%) (Fig 1c). Bioinformatics was done by "in house" analysis in 47 out of 55 centres.

NGS is the most frequent diagnostic approach, used for ≥50% of all genetic testing (Fig 2a) and in most centres (60%) NGS is offered to all patients with clinical criteria of NMD (Fig 2b). NGS follows traditional workup in 54% of participating centres (Fig 2c) and the most common way of patient referral to the laboratory is via specialists, either medical geneticists (42%) or neurologists (35%). An interesting modality of referral is via multidisciplinary teams, which appears quite common within the Euro-NMD centres (18%) (Fig 2d).

The majority of participating centres reported that VOUS are re-evaluated at defined time intervals (57%) (Fig 3a), while secondary findings are reported relatively rarely due to the predominant use of panel testing (75%) (Fig 3b). Routine validation of NGS results by Sanger sequencing is performed by 83.6% of laboratories, meaning that NGS output is still considered not accurate enough to skip technical validation (Fig 3c). 34 out of 55 centres do participate in external quality assessment schemes (as EMQN and EQA), either for routine testing or for NGS-based testing.

### Other aspects

The survey also explored some socio/economic aspects of the use of NGS. In general, the centers reported a quite good experience with NGS (94.5%—Fig 4a). The commonest barrier for the poor NGS implementation was lack of reimbursement (30.9% of centres) (Fig 4b).

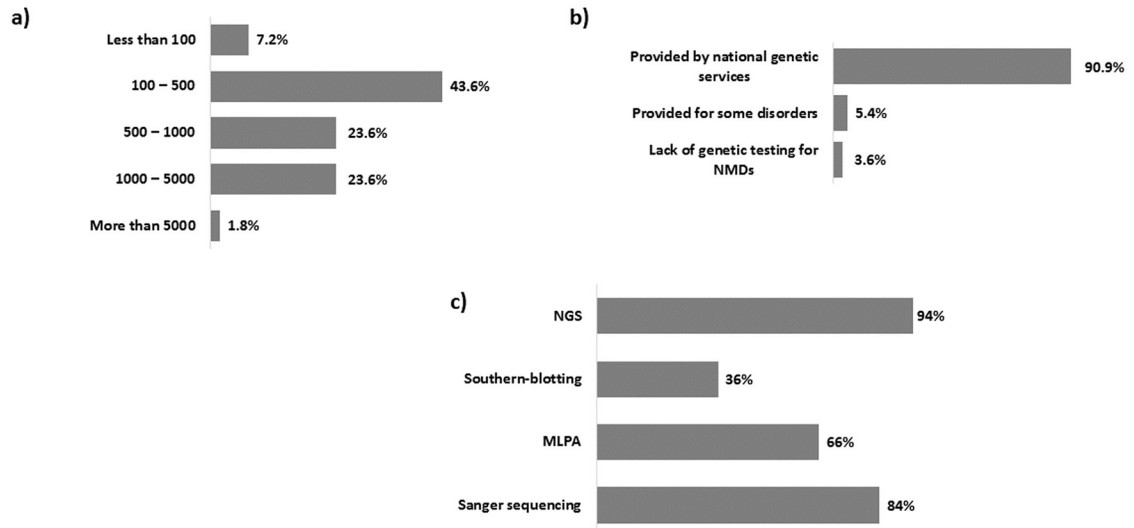

**Fig 1.** Availability of genetic test for NMDs in European Countries within Euro-NMD and most commonly used techniques: a) number of genetic tests (of all kinds) for NMDs performed in Euro-NMD HCPs; b) coverage of genetic tests for NMDs in Euro-HCPs; c) most commonly used technologies for NMDs diagnosis.

Availability of a database of NMD- associated variants was anticipated as the activity which could improve implementation of NGS in genetic testing of NMDs (85%), followed by NGS technical implementation (63%), standardisation (53%), and educational aspects to sensibilize clinicians (61%) (Fig 4c).

## EUGT Orphanet related data

Finally, based on the data exploited via the Orphanet database, since 81.8% (45 centres) of participants provided the EUGT code, we calculated how many centres are diagnosing the most frequent NMD diseases, with particular focus on some diseases for which personalized

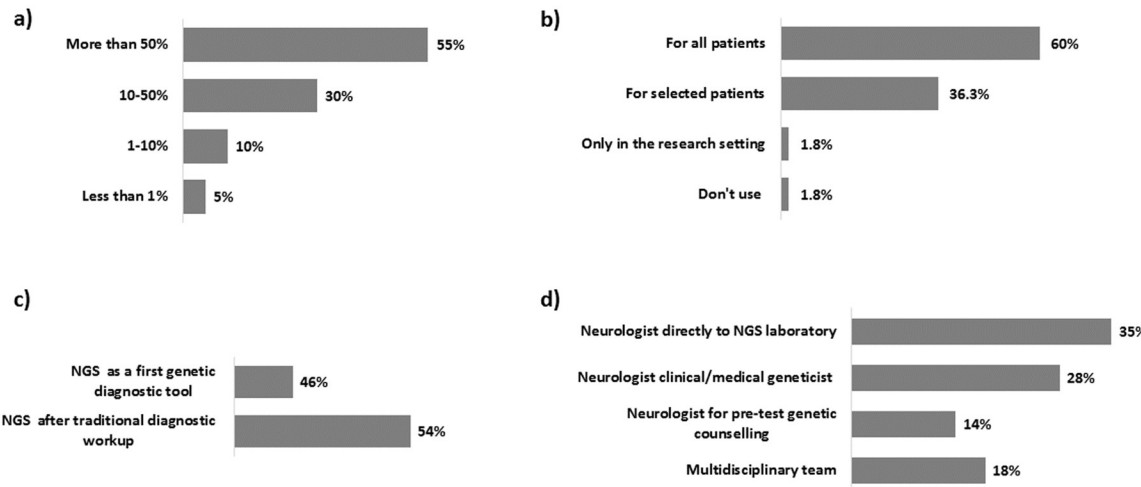

**Fig 2.** Impact of NGS in NMDs diagnosis and flowcharts for NGS testing: a) contribution of NGS to all genetic tests for rare NMDs in Euro-NMD HCPs; b) criteria for offering NGS approach; c) NGS prioritization in diagnostic algorithms; d) clinical pathway to refer patients for NGS testing.

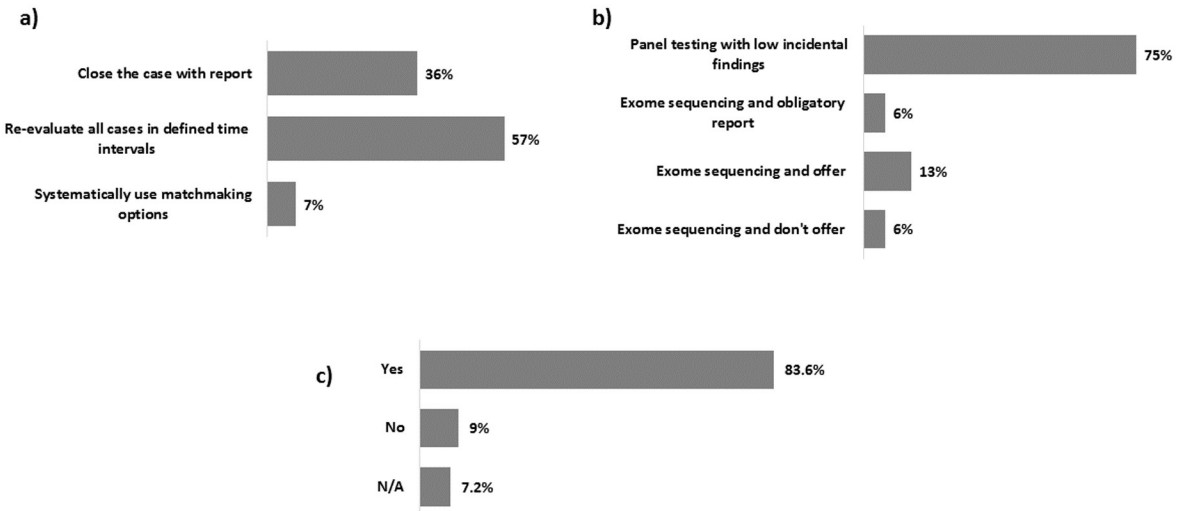

**Fig 3.** Euro-NMD centers strategies for reporting variants of uncertain significance (VOUS) and incidental findings and for validating NGS variants: a) centre's approach to VOUS; b) centre's approach to incidental findings; c) use of Sanger sequencing validation of NGS identified variants.

therapies are already available or in clinical trial phase. We selected spinal muscular atrophy (SMA), Duchenne muscular dystrophy (DMD), Pompe disease, Limb-girdle muscular dystrophies (LGMDs) and Transthyretin-related peripheral neuropathy. Fig 5 shows the results of these statistics.

**SMA.** SMN gene test (Fig 5a) was available in 23 centers (51.1%) with 33.3% of centres testing only *SMN1* gene, and 17.7% also testing *SMN2* gene copy numbers. Eight centres offer MLPA (for *SMN1* and *SMN2* copy number variation detection) only, 3 offer MLPA and sequencing (either Sanger or NGS) to identify small mutations and 10 centres did not specify the methods adopted.

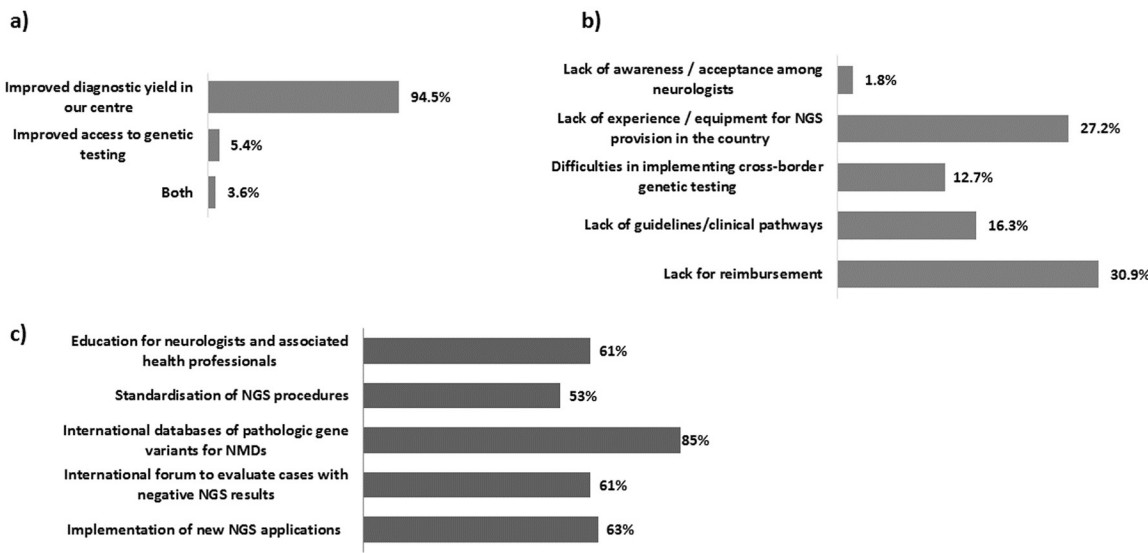

**Fig 4.** Social and economic aspects of NGS method application within Euro-NMD: a) NGS experience in Euro-NMD centers; b) experienced barriers for NGS implementation; c) activities needed to NGS implementation.

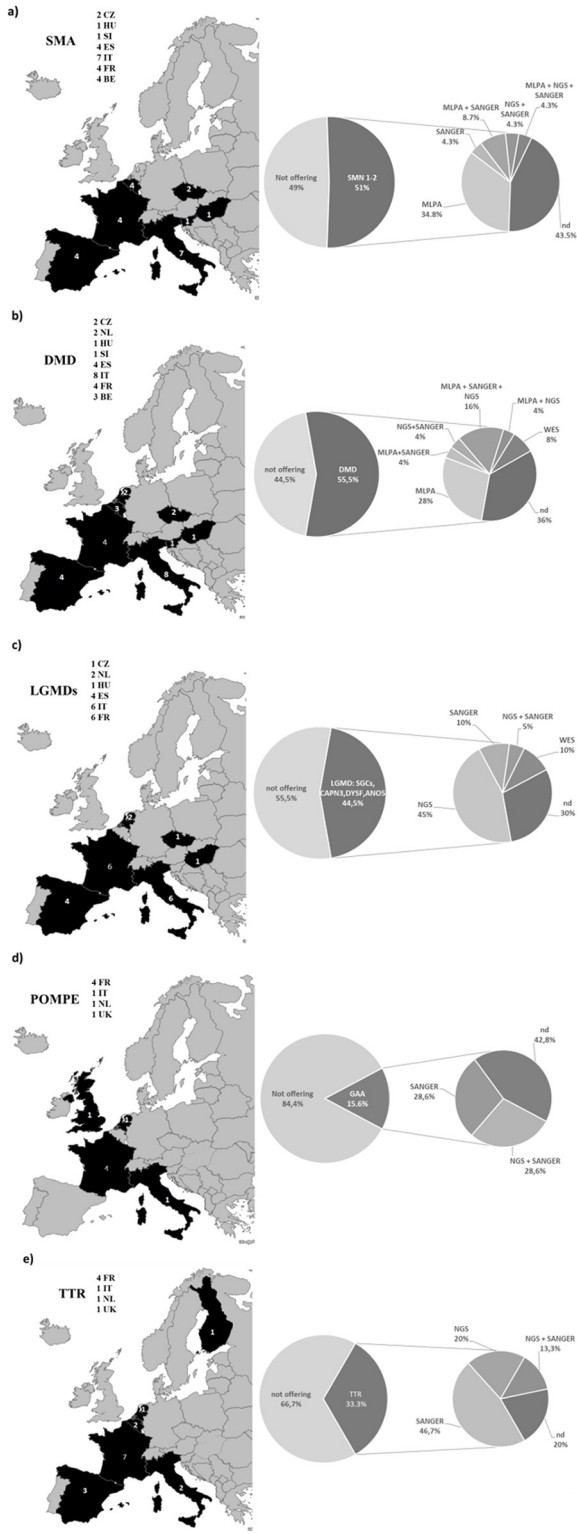

**Fig 5. Testing of selected RDs across Euro-NMD consortium centres providing unique EUGT number.** For each disease, the percentage of centres offering diagnosis, distribution of centres across European countries and technologies used are reported. a) SMN genes testing; b) DMD gene testing; c) LGMDs genes (SGCs: Sarcoglycan α, β, γ; CAPN3: Calpain3,DYSF: dysferlin; ANO5: anoctamin 5);d) GAA gene; e) TTR gene testing.

**DMD.** Centres offering DMD gene diagnostic (Fig 5b) were 25 (55.5%). Among these, 13 perform MLPA, 6 Sanger sequencing and 8 NGS, 2 centers have in place WES for DMD genetic testing, and 9 did not specify.

**LGMD.** LGMD is a very heterogeneous group of muscular dystrophies. In this analysis, we considered only the most frequent causative gene variants. A total of 20 centres (Fig 5c) performs genetic testing for LGMD genes (44.5%). Among these 13 test SGCA, 13 SGCB, 14 SGCG, 17 calpain-3, 15 dysferlin, 13 anoctamin, and 3 profile other LGMD genes.

The vast majority of centers use NGS (45%), WES (10%) and Sanger sequencing (10%) for genetic testing. No centres reported the use of MLPA or other methods to detect CNVs in the explored LGMD genes.

**POMPE disease.** Only a few Centers (7–15.6%) offer genetic testing for Pompe disease (Fig 5d). The methods used in 57.2% of centres were Sanger sequencing or NGS; 14,5% of Centers also test for analytes.

**TTR.** Although this is an ultrarare disease with a frequency less than 1:100,000, 15 centres (33.3%) offer TTR genetic diagnosis (Fig 5e). Used technology was specified for 12 centres. All of them use Sanger sequencing, which is preferred and cost-effective for the short TTR gene analysis. NGS was used in 3 centres, and both Sanger and NGS were used in 2. Three centres did not specify the methods adopted.

## Discussion

We provide here the results of the first survey launched within an ERN (Euro-NMD, https://ern-euro-nmd.eu) about the genetic testing service for NMD patients. We have also inquired more in detail about the use of NGS in a clinical diagnostic setting. The use of NGS is being progressively introduced in clinical diagnostic, and it represents an important diagnostic challenge for the NMD community.

### Response

Of the 61 centres, 55 from 12 different EU countries replied to the survey (90.2%). This is an excellent response, and we may infer that our collected answers cover the majority of the ERN genetic centres. A second reflection is that all centres are public institutions. In Europe, national health services have entirely different structures. We may have full public services (ex: UK, Italy, Spain, Sweden, Denmark, Norway, and Finland) which are based on the Beveridge model or mixed public/private services (ex: France, Germany, Austria, Switzerland, Belgium and The Netherlands) following the Bismarck model [10]. This might be related to the fact that care for rare diseases patients is complex, expensive and requires a certain degree of expertise that might be most accessible through public services, very often based on university hospitals. It might be interesting to monitor how the current situation may evolve in the future, as a consequence of enhance NGS d usage and availability. This window may deserve further attention for its social and economic implications.

### Number of genetic tests offered

The majority of centres (26/55; 47.2%) performed 500–5000 tests per year, while only 7.2% performed ≤ 500 tests and only one centre performed ≥ 5000 genetic tests per year (1.8%). The availability and the number of genetic tests performed across the ERN centres is very good, and we may envisage that whole-genome sequencing (WGS) implementation and translation into diagnostic pipelines will further increase the genetic test accuracy and offer availability. From this survey, we cannot ascertain if this number of tests cover the full range of genetically identifiable NMD since the methods in use are still various and include, besides

NGS, standard techniques such as gene-by-gene approaches. This represents an interesting research area deserving further investigation, but it was not in the scope of the survey. Nevertheless, by accessing the specific EUGT numbers, we have investigated this aspect for a few diseases (SMN, DMD, LGMDs, Pompe disease and TTR related neuropathy) as discussed below.

## Orphanet registration

Orphanet registration (www.orphanet.net) and the related EUGT number was reported by 81.8% of participating centres. Being registered in the most commonly used database for RDs is generally considered crucial information for the RD community as it harbours a considerable amount of vital information regarding RD diagnostics, care and therapy. The percentage of registered centres (81.8%) is significant; nevertheless, it needs to be improved since ideally all diagnostic centres should be reported in Orphanet to enhance transparency and maximize patient's access to appropriate and excellent care [11].

## NGS strategies, pipelines, VOUS and incidental findings reporting modalities

Although European tertiary genetic centres do still use a wide range of molecular biology techniques to address different mutational types, we have noticed a high rate of NGS implementation. In ≥ 50% of the EURO-NMD centres, NGS represents more than 50% of all genetic tests performed, while 15% of centres still predominantly use traditional approaches to molecular genetic diagnosis of NMD. The use of NGS was reported to be associated with increased diagnostic yield and/or improved accessibility to genetic testing by most of the centres. Similarly, several groups reported an increased yield in comparison with the gene-by-gene testing approach [4–6, 8].

Two-thirds (42/55) of centres used gene panel testing as NGS strategy and 13 centres used WES. Although, almost half of the participating centres already use NGS as a first-tier diagnostic tool. Most diagnostic guidelines for neuromuscular disorders predated the extensive use of NGS and therefore favoured genetic testing of a smaller number of candidate genes [12]. Recently, several studies indicated that early use of NGS might reduce the diagnostic odyssey characteristic for NMD [8, 13].

Clinical pathways for NGS differ significantly amongst the ERN centres, even within national health systems. Most frequently, the clinical specialists (neurologists, paediatric neurologists, clinical geneticists) send the patients' samples directly to the genetic laboratories (35%). Some centres involve multidisciplinary teams in the decision of when to prescribe NGS, while clinical geneticists play the central role in NGS testing prescription in 42% (in 14% of cases trough pre-test genetic counselling) of centres. The existence of different pathways might influence the yield of the testing. It has been shown that NGS centres use rather similar technologies and bioinformatics techniques; however, medical interpretation and reporting still differs among centres [14]. NGS testing is associated with several potential pitfalls; therefore, phenotype–driven NGS testing remains vital [15, 16]. Recent studies demonstrated that clinical specialists consider WGS beneficial for diagnosis, although it is still insufficiently spread across diagnostic genetic centres [17]. The ongoing ESHG and Eurogentest validation pathway to define the WGS diagnostic guidelines has the potential to make this approach more largely used in genetic testing practice (www.eurogentest.eu).

The ERN centres use different strategies to deal with VOUS. While more than one-third close the case by reporting the VOUS, the majority re-evaluate the finding at defined time intervals, and a minority (7%) use matchmaking to ascertain pathogenicity of VOUS. The strategy of revaluating VOUS at defined time intervals may be limited to predominantly

tertiary/ quarterly centres. A recent study stated that approximately one-third of diagnostic centres reported that reinterpretation of variants for clinical purposes might occur [18]. On the other hand, there is evidence that an ongoing variant interpretation is required to deliver the state of the art of genetic services [19]. Additionally, matchmaking has been shown to improve diagnostic efficiency for patients with rare genetic disorders [20].

Those centres using gene panel testing as an NGS diagnostic approach reported a low level of incidental findings. Centres, using exome sequencing offer optional reporting of the severe, actionable genetic conditions (13%). In contrast, in a minority of centres, reporting is obligatory. International consensus on reporting incidental findings in terms of actionable gene/diseases list and procedures is still developing [21–23].

## Barriers

The existence of barriers to NGS implementation was not reported frequently in our centres. The majority (31%) pointed out the lack of reimbursement as the main hurdle followed by lack of experience and lack of professional guidelines, 27.2% and 6.3% respectively. Reimbursement issue is not specific for NGS tests only as it has demonstrated already that there is considerable difference in access to genetic testing in EU, due to inadequate financing of genetic testing within the national health systems as well as in terms of cross-border provision of genetic testing [24]. Furthermore, introduction of NGS in the national health system is challenging not only in terms of financing but also due to the paucity of technical and clinical standards, reconfiguration of professional roles and need for establishing new service models [25]. Most of the expertise in diagnostics of rare, genetic disorders is concentrated in public medical tertiary centres and private initiatives do not play a major role in EU countries.

Limited evidence of benefit/value, acceptance among neurologists and difficulties in implementing cross-border genetic testing were not considered as significant barriers for NGS implementation in clinical use.

Among the activities that could potentially improve NGS implementation, EURO-NMD centres (85%) suggested the establishment of a database for pathogenic variants related to NMDs as the most useful, followed by new NGS applications including whole genome sequencing and RNA sequencing (63%). Other favoured activities were the establishment of a forum for evaluating patients were no pathologic variant was found (61%), education for neurologists (61%) and standardisation of NGS procedures (53%). The ERNs have available a unique platform provided by the EU, the Clinical Patient Management System (CPMS), which is probably still in its implementation phase in terms of ERN diffusion (see https://cpms-training.ern-net.eu/login/). Indeed, as soon as the CPMS is widely and daily used by HCPs, it will undoubtedly serve as a forum where patients' data can be shared and discussed, therefore enhancing the genetic test capacity, use, and interpretation.

Transcriptome sequencing [26, 27] and WGS [28] are promising strategies to improve genetic testing accuracy, but still in the clinical validation phase. On the other hand, education, and standardisation of NGS procedures are activities that may, in a short time frame contribute to the wider implementation of NGS.

Bioinformatics was done by "in house" analysis in 47 out of 61 centres, being a quite common modality and showing the specialism of the ERN centres. Finally, but quite important, 34 out of 61 centres do participate yearly in external quality assessment schemes such as EMQN and EQA, for routine genetic testing and NGS-based testing. The participation in external quality assessment schemes ensures quality control to genetic testing and consequently guarantees an accurate genetic diagnosis.

When analysing these answers, it is important to know that the genetic laboratories that participated in this Survey are all part of the ERN Euro-NMD. The healthcare providers that belong to EURO-NMD are all public institutions with both research and diagnostic activities. For the purpose of the Survey, we were mostly interested in the diagnostic activities. To be part of the ERN the healthcare providers had to fulfil specific criteria namely to be able to provide genetic testing for the rare diseases they have expertise on. The funding and reimbursement rules for the different centres are particular to each health-care system as in Europe health policies are a matter of each country. Most European countries apply cost containment measures due to the high costs of health care. These go from expenditure ceilings, limits on human resources, limits on the availability of certain technologies, policies of cost sharing and exclusion of certain tests from coverage. [http://www.oecd.org/science/emerging-tech/34779945.pdf).

## Databases

The use of databases for managing NMD data is considered a useful tool for both clinicians and researchers. However, the number of existing databases, the incompleteness of data sets and the inoperability amongst these make the task of variant validation for clinical use cumbersome. Locus-specific databases (LSDBs) aim to collect all published and unpublished (pathogenic) variants for a specific gene along with complete clinical and phenotypic information. They are usually trustworthy as they are curated. There are over 700 LSDB that vary in their content, completeness, time available for curation, and the expertise of the curator. This spread in the number of databases led to the creation of a federation of Locus-Specific Database (LSDB) curators [29] with the subsequent establishment of the Mutation Database Initiative (MDI) under the auspices of the Human Genome Organisation (HUGO); MDI later became a society, the Human Genome Variation Society (HGVS) http://www.hgvs.org/entry.html.

Guidelines to drive the creation of LSDB are in place. They address, in particular, what data to collect and how-to curate data. However, not all databases take into consideration the FAIR data principles or the use of ontologies, two significant aspects that should be considered when addressing the interoperability of databases and data sharing. The FAIR Data Principles are a set of guiding principles that enable data to become Findable, Accessible, Interoperable and Reusable [30]. Ontologies are controlled vocabularies with a systematic structure that allows easy data retrieval and analysis. The Human Phenotype Ontology (HPO) http://www.human-phenotype-ontology.org/) (is an ontology developed for describing phenotypic abnormalities. For describing the effects of variation on DNA, RNA, protein sequence, structure, function, interaction, and other features, the Variation Ontology (VariO) (http://variationontology.org/) is a useful tool. Equally important is the variant annotation that should follow standard methods such as the Sequence Variant Nomenclature recommended by the Human Genome Variation Society (HGVS). EURO-NMD was recently awarded a grant (Grant Agreement number: 947598—EURO-NMD Registry—HP-PJ-2019) to establish the EURO-NMD Core Registry Hub. This registry will follow interoperability guidelines being developed within the European Joint Program for Rare Diseases (EJP RD). It will closely work with EJP RD data stewards to implement these guidelines in the EURO-NMD existing registries. The aim is to showcase a model that will set the golden standard for out-of- silo thinking and data sharing.

## Disease-specific data

Since we had the EUGT code of 82% of centres, we analysed the genetic testing service for five key-RDs, for which personalized therapies, and consequently therapeutic windows, are available.

The genetic approach to diagnose SMA is quite simple and based on MLPA (covering 98% of deletion mutations and also identifying *SMN1* and *SMN2* copy number). SMA testing was routinely offered in 23 centers. Nevertheless, we observed that only 17,7% of centres also include in the diagnostic report *SMN2* gene copy numbers counting, being this test technically challenging and not strictly required for SMA genetic confirmation. As a consequence of the very recent approval of Spinraza for SMA, detecting *SMN2* copy number is now unavoidable. Centers migth/should be organizing in order to implement this aspect of SMA genetic diagnosis [3]. It has to be underlined that SMA is a relatively frequent disease; therefore, a more widespread availability of genetic testing might be desirable and probably needed. The introduction in some countries of newborn screening for this disease may eventually provide an early genetic identification [31].

**DMD.**  Vast majority (13) of centers performed MLPA, and 9 provided sequencing, using also WES (2 centres). DMD is a RD with a frequency of 1/5,000 born males. Therefore, also for this case, the genetic testing service seems low, and it may need to be increased. Considering that the only (provisionally) approved orphan drug in Europe is Translarna, the need for extensive diagnosis by identifying all mutation types (including small mutations) is vital for being eligible for this therapy. Accurate testing for DMD will also be even more relevant when antisense oligoribonucleotide-based therapy and gene therapy become available [32]. We conclude that also for DMD the diagnostic need is not fully met.

**LGMD.**  Only a few LGMD genes were regularly tested (*SGCA*, *SGCB*, *SGCG*, *CAPN3*, *DYSF*, *ANO5*) by NGS gene panels or rarely WES (2 centers only). LGMDs seem to be underdiagnosed and will probably benefit from the use of WGS because of its high genetic and clinical heterogeneity, with WGS being able to detect small mutations and copy number variation (CNVs). Also, there was a complete lack of use of MLPA to detect deletions and duplication in the LGMD genes, and this reduces the detection rate of the diagnostic pipeline. An accurate and high detection rate for all LGMD genes will be vital as soon as new therapies are available.

**POMPE disease.**  Pompe disease genetic testing is offered by only 7 centers. This is not surprisingly since it is an ultrarare disease (1:60,000 (late onset)-1:100,000 (infantile onset). Another explanation for this very limited offer might be that the companies providing the drugs for enzyme replacement therapy (Sanofi Genzyme https://www.sanofigenzyme.com/; Amicus Therapeutics https://www.amicusrx.com/), in some countries, also cover the costs of the genetic test. This is a private, free of charge service that might become more common in the future when other drugs for very- or ultra-rare diseases become available. This model might offer interesting opportunities for public and private systems to interact to improve patients' care synergically. It may also raise ethical questions if not correctly planned. Items such as which laboratories are allowed to perform these tests, who has access to the results, and who is allowed to request such tests, should be thoroughly discussed.

**TTR.**  For this ultrarare disease (1:100,000), 15 centres offered TTR genetic diagnosis. This is a quite relevant number (twice compared to Pompe disease) that is possibly related to the recent availability of personalised therapy (TTR gene silencing); [33] for TTR-related amyloidosis together with the small size of the gene (4 exons).

Although the survey data may suggest that some genetic test implementation may be needed, for some countries genetic tests for a specific disease are heavily centralized. The extent of this test offer organization in various countries is however not obtainable by this survey.

## Conclusions

The results of this study should be treated cautiously in terms of generalisation of the findings since it applies only to the ERN Euro-NMD members. Nevertheless, the response rate (90.2%)

makes our data a good snapshot of the genetic testing landscape for NMD in Europe. Consequently, since the survey was completed by centres of expertise belonging to the ERN, it might be expected that the level of implementation, barriers and needs in secondary genetic centres might differ from those reported in this study.

In conclusion, the number of genetic tests offered within the ERN is very high; current genetic diagnostic techniques across countries are dissimilar in terms of pipelines and percentages, but quite homogenous in terms of types. This is especially true for NGS approaches, which are widely adopted within Euro-NMD. Indeed, the use of NGS techniques is part of the routine testing for NMD patients. However, barriers still exist due to lack of standardisation of methods, lack of expertise and education, and social and economic issues related mostly with reimbursement. This last aspect appears to be challenging to resolve, due to the country-specific national health system organization.

Said that ERNs, as Euro-NMD, may be an excellent educational and information instrument, to help national policymakers to better understand the needs of the RD community.

We, therefore, believe that the Euro-NMD survey was able to point the gaps in the current practice, to draft a realistic map of the genetic testing modalities and offers, and it may help to delineate new strategies at the national level to implement genetic testing for NMD access and availability, so that wherever patients are, they can have equitable access to diagnosis.

## Supporting information

**S1 Table. Survey questionnaire.**
(PDF)

**S2 Table. List of centres participating in the survey and Orphanet EUGT numbers.**
(XLSX)

## Acknowledgments

All the authors of this publication are members of the European Reference Network for Rare Neuromuscular Diseases (EURO-NMD) Project ID No 739543. SS, SM ad BP and AF (Chair) are constituents of the Genetic Task. Alexander Mejat and Francoise Lamy (French Muscular Dystrophy Association AFM-Téléthon) are kindly acknowledged as members of the EURO-NMD Genetic Task.

## Author Contributions

**Conceptualization:** Borut Peterlin, Teresinha Evangelista, Alessandra Ferlini.

**Data curation:** Borut Peterlin, Francesca Gualandi, Ales Maver, Serenella Servidei, Silvère M. van der Maarel, Francoise Lamy, Alexander Mejat, Teresinha Evangelista, Alessandra Ferlini.

**Formal analysis:** Borut Peterlin, Francesca Gualandi, Ales Maver, Serenella Servidei, Silvère M. van der Maarel, Francoise Lamy, Alexander Mejat, Teresinha Evangelista, Alessandra Ferlini.

**Writing – original draft:** Borut Peterlin, Francesca Gualandi, Ales Maver, Serenella Servidei, Silvère M. van der Maarel, Francoise Lamy, Alexander Mejat, Teresinha Evangelista, Alessandra Ferlini.

**Writing – review & editing:** Borut Peterlin, Teresinha Evangelista, Alessandra Ferlini.

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
