## [Decision Letter · Decision Letter 0]

28 Jul 2020

PONE-D-20-14793

Genetic testing offer for inherited neuromuscular diseases within the EURO-NMD reference network: a European survey study.

PLOS ONE

Dear Dr. Ferlini,

Thank you for submitting your manuscript to PLOS ONE. After careful consideration, we feel that it has merit but does not fully meet PLOS ONE’s publication criteria as it currently stands. Therefore, we invite you to submit a revised version of the manuscript that addresses the points raised during the review process.

The single expert reviewer felt that your paper was well written and that your finds support your conclusions. This reviewer did make useful suggestions for its improvement.

We look forward to receiving your revised manuscript.

Kind regards,

Alfred S Lewin, Ph.D.

Academic Editor

PLOS ONE

Journal Requirements:

Additional Editor Comments (if provided):

Reviewers' comments:

Reviewer's Responses to Questions

**Comments to the Author**

1. Is the manuscript technically sound, and do the data support the conclusions?

Reviewer #1: Yes

2. Has the statistical analysis been performed appropriately and rigorously? 

Reviewer #1: Yes

3. Have the authors made all data underlying the findings in their manuscript fully available?

Reviewer #1: Yes

4. Is the manuscript presented in an intelligible fashion and written in standard English?

Reviewer #1: Yes

5. Review Comments to the Author

Reviewer #1: This is an interesting report that outlines the genetic diagnostics of inherited neuromuscular diseases within the EURO-NMD European Reference Network (ERN). The results are interesting but not unexpected. I have a few comments.

1. In the Abstract, in the first paragraph, the authors talk about the collection of “information about the diffusion/distribution of genetic testing across 61 ERN healthcare providers.” It is not clear to me what the authors mean by “diffusion” of genetic testing. They probably mean the “availability” of genetic testing, but I am not sure. Using a different word will make it clear to the readers of this paper.

2. In the Introduction, line 91, the statement about lack of reimbursement is interesting given that all the surveyed centers providing the testing were public institutions. You would expect a government-run healthcare system to cover the expenses. I think this is an interesting topic, and hence it might be useful for the authors to elaborate on that and to compare it, for example, to private institutions. It would also be interesting to know the impact of lack of reimbursement on the capacity of these laboratories to offer the testing ordered by clinicians.

3. The paper is well written; however, it feels as if sometimes the information is repeated. It might be useful to avoid repetition in the paper. For example, in the Results section and under EUGT Orphanet Related Data, the authors have selected a number of diseases such as SMA, DMD, Pompe’s disease, limb-girdle muscular dystrophies, and transthyretin related peripheral neuropathy, to describe specific data about these diseases. They do the same in the Discussion section of the paper in more detail. My advice would be to present the information only in one section, perhaps the Results section, and make a few comments in the Discussion about this group of diseases.

4. The Barriers to Testing section is very interesting. Again, the lack of reimbursement seems to be a main hurdle, followed by lack of experience and lack of professional guidelines. It might be interesting for the readers to know which are the testing sites within ERN. It is not clear from this paper whether there is a central agency that certifies these laboratories, as happens in other countries, like the USA. Are these clinical or research-based labs?

5. In the section under Databases, the authors do not list commonly used databases in other parts of the world, like the Exome Variant Server database, the NIH SNP database, and others, and I do not understand the reason for that.

6. Under Disease Specific Data, in the SMA paragraph, the authors observe that only 17.7% of the centers also tested SMN2 gene copy number. Is this because of lack of technical expertise? It is not clear.

6. PLOS authors have the option to publish the peer review history of their article (what does this mean?). If published, this will include your full peer review and any attached files.

Reviewer #1: No

---

## [Author Response · Author response to Decision Letter 0]

1 Sep 2020

Review Comments to the Author

Reviewer #1: This is an interesting report that outlines the genetic diagnostics of inherited neuromuscular diseases within the EURO-NMD European Reference Network (ERN). The results are interesting but not unexpected. I have a few comments.

1. In the Abstract, in the first paragraph, the authors talk about the collection of “information about the diffusion/distribution of genetic testing across 61 ERN healthcare providers.” It is not clear to me what the authors mean by “diffusion” of genetic testing. They probably mean the “availability” of genetic testing, but I am not sure. Using a different word will make it clear to the readers of this paper.

We thank the reviewer for this comment. It is right, we meant availability, we have rephrased the sentence.

2. In the Introduction, line 91, the statement about lack of reimbursement is interesting given that all the surveyed centers providing the testing were public institutions. You would expect a government-run healthcare system to cover the expenses. I think this is an interesting topic, and hence it might be useful for the authors to elaborate on that and to compare it, for example, to private institutions. It would also be interesting to know the impact of lack of reimbursement on the capacity of these laboratories to offer the testing ordered by clinicians.

In line with the reviewer’s suggestion we elaborated on the financing issues related to NGS testing by expanding the concept in the Introduction and implementing Discussion.

3. The paper is well written; however, it feels as if sometimes the information is repeated. It might be useful to avoid repetition in the paper. For example, in the Results section and under EUGT Orphanet Related Data, the authors have selected a number of diseases such as SMA, DMD, Pompe’s disease, limb-girdle muscular dystrophies, and transthyretin related peripheral neuropathy, to describe specific data about these diseases. They do the same in the Discussion section of the paper in more detail. My advice would be to present the information only in one section, perhaps the Results section, and make a few comments in the Discussion about this group of diseases.

We agree with the reviewer and we amended that accordingly with shorten and rephrased paragraphs.

4. The Barriers to Testing section is very interesting. Again, the lack of reimbursement seems to be a main hurdle, followed by lack of experience and lack of professional guidelines. It might be interesting for the readers to know which are the testing sites within ERN. It is not clear from this paper whether there is a central agency that certifies these laboratories, as happens in other countries, like the USA. Are these clinical or research-based labs?

Thanks for this comment; we have added a sentence specifying what the referee requested.

5. In the section under Databases, the authors do not list commonly used databases in other parts of the world, like the Exome Variant Server database, the NIH SNP database, and others, and I do not understand the reason for that.

We thank the reviewer for his comment. The authors were trying to define the problems of the existing databases and the diversity of data that is collected in each one. By no means had we wanted to exclude databases in other regions of the world, we have tried to use examples that were best known to us and tools that in the last years have received wide approval in Europe such as the HPO or the FAIR principles.

6. Under Disease Specific Data, in the SMA paragraph, the authors observe that only 17.7% of the centers also tested SMN2 gene copy number. Is this because of lack of technical expertise? It is not clear.

The referee is right the sentence was not clear. We have modified the SMA paragraphs.

---

## [Editor Report · Decision Letter 1]

4 Sep 2020

Genetic testing offer for inherited neuromuscular diseases within the EURO-NMD reference network: a European survey study.

PONE-D-20-14793R1

Dear Dr. Ferlini,

We’re pleased to inform you that your manuscript has been judged scientifically suitable for publication and will be formally accepted for publication once it meets all outstanding technical requirements.

Kind regards,

Alfred S Lewin, Ph.D.

Section Editor

PLOS ONE
---

## [Editor Report · Acceptance letter]

9 Sep 2020

PONE-D-20-14793R1

Genetic testing offer for inherited neuromuscular diseases within the EURO-NMD reference network: a European survey study.

Dear Dr. Ferlini:

I'm pleased to inform you that your manuscript has been deemed suitable for publication in PLOS ONE. Congratulations! Your manuscript is now with our production department.

Kind regards,

on behalf of

Dr. Alfred S Lewin 

Section Editor

PLOS ONE